# YOLOv8-DEE: a high-precision model for printed circuit board defect detection

Feifan Yi[1,2], Ahmad Sufril Azlan Mohamed[1], Mohd Halim Mohd Noor[1], Fakhrozi Che Ani[3] and Zol Effendi Zolkefli[3]

[1] School of Computer Sciences, Universiti Sains Malaysia, Penang, Malaysia
[2] Institute of Applied Artificial Intelligence of the Guangdong-Hong Kong-Macao Greater Bay Area, Shenzhen Polytechnic University, Shenzhen, China
[3] Department of Manufacturing Engineering, Western Digital SanDisk Storage Malaysia, Penang, Malaysia



## ABSTRACT

Defects in printed circuit boards (PCBs) occurring during the production process of consumer electronic products can have a substantial impact on product quality, compromising both stability and reliability. Despite considerable efforts in PCB defect inspection, current detection models struggle with accuracy due to complex backgrounds and multi-scale characteristics of PCB defects. This article introduces a novel network, YOLOv8-DSC-EMA-EIoU (YOLOv8-DEE), to address these challenges by enhancing the YOLOv8-L model. Firstly, an improved backbone network incorporating depthwise separable convolution (DSC) modules is designed to enhance the network's ability to extract PCB defect features. Secondly, an efficient multi-scale attention (EMA) module is introduced in the network's neck to improve contextual information interaction within complex PCB images. Lastly, the original complete intersection over union (CIoU) is replaced with efficient intersection over union (EIoU) to better highlight defect locations and accommodate varying sizes and aspect ratios, thereby enhancing detection accuracy. Experimental results show that YOLOv8-DEE achieves a mean average precision (mAP) of 97.5% and 98.7% on the HRIPCB and DeepPCB datasets, respectively, improving by 2.5% and 0.7% compared to YOLOv8-L. Additionally, YOLOv8-DEE outperforms other state-of-the-art methods in defect detection, demonstrating significant improvements in detecting small, medium, and large PCB defects.

## INTRODUCTION

Quality control is crucial in the PCB manufacturing process (*Wang et al., 2022*). PCB defects can generally be classified into two categories: functional defects and cosmetic defects (*Wu, Wang & Liu, 1996*). Functional defects, such as component damage or missing, critically impact the PCB's performance and can render it inoperable for end users. These defects are typically identified through X-ray inspection or power-on testing. On the other hand, cosmetic or appearance-related defects encompass issues such as open, short, mousebite, spur, pin-hole, spur. While these primarily affect the visual quality of the PCB, they can also escalate into functional failures over time with prolonged use. Effectively detecting appearance defects on PCBs has been a key focus for researchers in

Corresponding author
Ahmad Sufril Azlan Mohamed, sufril@usm.my

both industry and academia. Initially, PCB inspection relied on manual methods. However, manual detection is inefficient and prone to errors, making it inadequate for industrial production demands (*Yang et al., 2020*).

With advancements in computer vision field, automated optical inspection (AOI) methods based on computer image processing technology have been increasingly adopted by PCB manufacturers (*Li & Yang, 2011*). AOI systems utilize industrial cameras with optical lenses to capture images, applying various image processing algorithms such as edge extraction (*Leek et al., 2006*), corner detection (*Rosten & Drummond, 2006*), and line detection (*Aggarwal & Karl, 2006*) to analyze these images. These algorithms help identify the location and type of PCB defects. Despite the effectiveness of AOI technology in specific applications, its generalization remains challenging. Variations in light intensity, product types, and background environments often necessitate redesigning and debugging established PCB algorithms, which is both time-consuming and costly for manufacturers.

The development of artificial intelligence offers new solutions for PCB defect detection. Data-driven deep learning methods utilize convolutional neural networks (CNNs) to extract high-dimensional features from images, effectively addressing the generalization and robustness issues inherent in traditional image processing methods. Currently, deep learning-based object detection algorithms are mainly categorized into one-stage and two-stage detection models. In two-stage approaches like Mask R-CNN (*He et al., 2017*) and Faster R-CNN (*Ren et al., 2016*), the process is divided into two distinct phases. First, the algorithm generates region proposals—a set of candidate regions in the image where objects might be located. These proposals are typically obtained using a Region Proposal Network (RPN). In the second stage, each candidate region is classified, and its bounding box is further refined. Two-stage models are known for their high accuracy due to the refinement step, making them more suitable for applications where precision is critical. However, this comes at the cost of increased computational complexity and slower inference speed, as the detection process involves multiple steps. On the other hand, one-stage approaches, such as SSD (*Liu et al., 2016*) and the YOLO (You Only Look Once) series (*Redmon et al., 2016*), aim to simplify the detection process by treating object detection as a single regression problem. These models directly predict the bounding box coordinates and object class probabilities in a single step without the intermediate region proposal stage. As a result, one-stage models are significantly faster, making them more suitable for real-time applications. However, this speed often comes with a trade-off in accuracy, especially for detecting small objects or objects in densely populated scenes, where the lack of a refinement stage can lead to missed or imprecise detections. The YOLO series, as a widely recognized one-stage detection model, excels in detecting defects of various sizes and types with superior speed compared to two-stage algorithms, achieving a better balance between detection speed and accuracy. Many researchers have successfully deployed YOLO algorithms in PCB quality inspection, achieving relatively better detection accuracy, which is significantly improved compared to existing methods (*Santoso et al., 2022*; *Mamidi, Sameer & Bayana, 2022*; *Tang et al., 2023*).

Although considerable progress has been made in PCB defect detection using CNN-based deep learning methods, the field still faces significant challenges:

1) Small target detection. Some tiny PCB defects are often only a few hundred pixels in size, which only account for just 0.005–0.07% in a high-resolution PCB image with a resolution of millions of pixels (*Yang et al., 2023*). Detecting defects of this scale presents significant challenges.

2) Multi-scale detection. The complex PCB manufacturing process is prone to a wide variety of defects with significant differences in scale. Small defects such as spurs are often less than 300 pixels, while large defects such as missing components may exceed hundreds of thousands of pixels.

3) Complex background. Highly integrated PCBs contain numerous components, text, soldering oil, and wire holes unrelated to defects. These backgrounds can obscure defect characteristics.

4) Precision. Even a small number of defects, or those at the micrometer scale, can result in a total loss of PCB functionality. Unlike other inspection tasks, where overall model accuracy is often the primary focus, PCB defect detection requires not only high overall accuracy but also emphasizes the precision for each individual defect category. This is because every type of defect, regardless of its frequency or size, has the potential to cause significant damage to the PCB and impair its functionality.

5) Real-time detection. High-resolution PCB images significantly increase computational time for defect detection algorithms. Balancing speed and accuracy with effective lightweight algorithms by reducing computational complexity and resource demands is crucial.

To address the second, third and fourth challenges of PCB defect detection tasks, this article proposes a new PCB defect detection algorithm, which enhances the feature extraction, feature fusion, and target detection capabilities of the YOLOv8-L model (*Jocher, Chaurasia & Qiu, 2023*). The main contributions of this article are as follows:

1) **Novel YOLOv8-DSC-EMA-EIoU (YOLOv8-DEE) framework**: An improved YOLOv8-L framework that achieves high detection accuracy across various indicators in PCB defect detection tasks.

2) **Enhanced backbone with depthwise separable convolution (DSC)**: The integration of DSC into the model's backbone significantly improves feature extraction capabilities while minimally increasing model complexity.

3) **Enhanced neck with efficient multi-scale attention (ema) module**: The neck of the model employs the EMA module to enhance feature fusion, improving information exchange between different feature layers and aiding in detecting small differences between various PCB defect types.

4) **EIoU as the regression loss function**: Utilizing EIoU during the training phase significantly improves prediction regression accuracy, enhancing positioning precision.

The rest of this article is organized as follows. A review of state-of-the-art (SOTA) PCB defect detection methods is presented in "Related Work". The details and improvements of the proposed YOLOv8-DEE model are illustrated in "Methodology". In "Experiment and Results", the proposed algorithm is validated on public datasets and compared with the performance of other algorithm models. Finally, "Conclusions" summarizes and analyzes the proposed method and considers directions for future work.

## RELATED WORK

### Traditional machine learning based methods for PCB defect detection

PCB defect detection algorithms based on machine learning typically integrate image processing techniques, such as template-based methods (*Zhou et al., 2023*). These methods employ image matching or subtraction to identify defect characteristics, which are then analyzed using machine learning algorithms (*Li et al., 2020*). *Crispin & Rankov (2007)* propose a method for automated PCB component inspection using a genetic algorithm with template matching. This approach tackles challenges like locating and identifying multiple components on a PCB image. It employs a normalized cross-correlation for template matching and a genetic algorithm to optimize the search process, reducing computational cost compared to exhaustive search. The results demonstrate the effectiveness of this method in detecting component placement errors on PCBs. Further, *Wu et al. (2013)* present a two-stage system for classifying solder joints in electronic devices. First, features are extracted from images, likely including color and shape properties. Then, a Bayes classifier efficiently separates good joints from defective ones. Finally, for defective joints, a support vector machine differentiates between various defect types. This method improves classification accuracy and efficiency by reducing the number of features needed for defect type identification. Another method, explored by *Hao et al. (2013)*, combines a neural network's ability to learn complex defect patterns with a genetic algorithm's optimization capabilities for automated solder joint inspection on printed circuit boards. This approach aims to improve the accuracy and efficiency of solder joint defect detection compared to traditional methods.

However, a major challenge with template-based methods is obtaining perfect template images. Different PCB inspection tasks require distinct template images, resulting in significant time and cost inefficiencies. Additionally, most machine learning-based PCB inspection algorithms are limited to defect classification, identifying the type of defect but not its location. This limitation arises because these algorithms function primarily as classifiers without positional capabilities.

Another significant traditional method is similarity measure. *Annaby, Fouda & Rushdi (2019)* propose an improved normalized cross-correlation (NCC) method for defect detection in PCBs. The authors enhance the traditional NCC by incorporating image preprocessing techniques, such as image normalization and noise reduction, to increase accuracy in detecting defects. They apply a sliding window approach to measure the similarity between the reference and test images pixel by pixel, using the enhanced NCC to identify mismatches that indicate defects. However, this method for defect detection is computationally intensive due to its pixel-by-pixel comparison and sliding window

approach, making it slower for high-resolution images. It also struggles with variations in lighting or alignment, reducing its robustness in uncontrolled environments.

To address these issue, deep learning algorithms offer a promising solution. Deep learning networks can extract high-dimensional features from images, enabling both defect classification and localization. Thus, the application of deep learning algorithms to PCB defect detection is likely to be the future trend.

## Deep learning based methods for PCB defect detection

In recent years, deep learning-based methods, particularly CNNs, have been widely adopted for PCB defect detection (*Yuan et al., 2024*; *Ding et al., 2019*). Unlike traditional machine learning methods, CNN-based approaches automatically extract image features and streamline the preprocessing process, thereby enhancing detection accuracy and speed. Some studies are devoted to solving the complex background problem in defect detection tasks to achieve better detection performance. For example, *Li et al. (2022)* propose a method to addresses challenges in industrial defect detection, particularly for small defects with unclear foreground-background separation. It builds upon the YOLO object detection framework but incorporates an "expanded field of feeling" to capture a wider range of contextual information. This method also utilizes feature fusion to combine different levels of detail from the image, enhancing defect characterization. The study demonstrates that YOLO-RFF achieves high-speed and accurate detection of various industrial defects.

In order to improve the detection accuracy of the model, some studies have achieved this by modifying specific convolution modules or adding attention mechanisms in the baseline model. A model called YOLO-MBBi is proposed by *Du et al. (2023)* for detecting surface defects on PCBs using an improved YOLOv5 deep learning model. Authors design this model to addresses limitations of YOLOv5 in this task, such as lower accuracy and speed. It incorporates various enhancements like MBConv modules and CBAM attention for faster and more precise defect recognition. The results show significant improvement in both accuracy (over 3% higher $AP_{0.5}$) and processing speed (achieving near 49 FPS) compared to the original YOLOv5, making YOLO-MBBi a promising solution for real-time PCB inspection. In the road defect detection task, *Wang et al. (2024)* introduced C2f-Faster-EMA and SimSPPF to replace the original C2f and SPPF modules in YOLOv8s, thereby improving the representation ability of the model and achieving higher accuracy on related data sets.

Moreover, some studies improve the performance by solving the multi-scale problem among defects. For example, *Wang et al. (2023)* introduces a novel multi-scale module to enhance YOLO-v5 for steel surface defect detection. This new module captures features at different scales more effectively by incorporating a more robust feature pyramid network, allowing better detection of small and complex defects. It improves the model's ability to handle varying defect sizes by extracting and merging multi-scale features through adaptive convolutional layers. This results in finer feature representation and greater accuracy, particularly for subtle defects, while maintaining real-time performance. The

multi-scale module significantly outperforms the standard version in defect detection tasks on steel surfaces.

# METHODOLOGY

Building on the success of previous YOLO models, YOLOv8 (*Jocher, Chaurasia & Qiu, 2023*) has gained popularity for various object detection tasks. YOLOv8 improves upon YOLOv5 (*Jocher, 2020*), achieving superior results across multiple detection tasks. As a single-stage object detection framework, YOLOv8 comprises three components: Backbone, Neck, and Head. The backbone extracts features from the input image, the neck processes feature information from different convolutional layers, and the Head predicts the object's category and location. Specifically, YOLOv8's backbone adopts the cross stage partial (CSP) network structure, which includes five feature extraction blocks {P1, P2, P3, P4, P5}. It replaces YOLOv5's C3 module with the C2f module, making the network more lightweight while maintaining feature extraction capabilities. The neck uses the PANet method based on the feature pyramid network (FPN) architecture and also replaces the original C3 module with the CSP with focus (C2f) module. In the head, YOLOv8 transitions from the previous anchor-based method to an anchor-free approach with a decoupled detection head.

## Overall framework of YOLOv8-DEE

Although YOLOv8 has demonstrated impressive results on the COCO dataset (*Lin et al., 2014*), recent studies have shown its generalization potential across various application domains. For instance, YOLOv8 has been successfully applied to insulator defect detection (*Jiang, Hou & Wang, 2024*), battery inspection (*Tzelepakis et al., 2023*), and steel surface inspection (*Kong & You, 2024*). These findings further support the network's versatility and robustness in different industrial contexts. In PCB defect detection, defect shapes and textures, such as spurs and open circuits, are often less distinct compared to typical images. Therefore, developing a high-precision PCB defect detection model based on YOLOv8 is crucial. This study proposes a novel detection model for PCB defects, termed YOLOv8-DEE. Figure 1 illustrates the YOLOv8-DEE framework. Based on YOLOv8-L, this model incorporates an enhanced backbone network using the DSC module and an improved neck based on the EMA module. Additionally, an efficient intersection over union (EIoU) loss function for bounding box regression replaces the original complete intersection over union (CIoU) loss in the head.

## Enhanced backbone with DSC module

In PCB defect detection, many areas are defect-free, and defects often exhibit only slight differences in texture, shape, and size compared to defect-free regions. This subtlety poses a significant challenge to the feature extraction capabilities of detection models. To address this issue, an improved backbone incorporating the DSC module (*Chollet, 2017*) is proposed in this article, which enhances the model's feature extraction ability.

As shown in Fig. 2, DSC module comprises two components: depthwise convolution and pointwise convolution. Depthwise convolution applies a separate convolution kernel

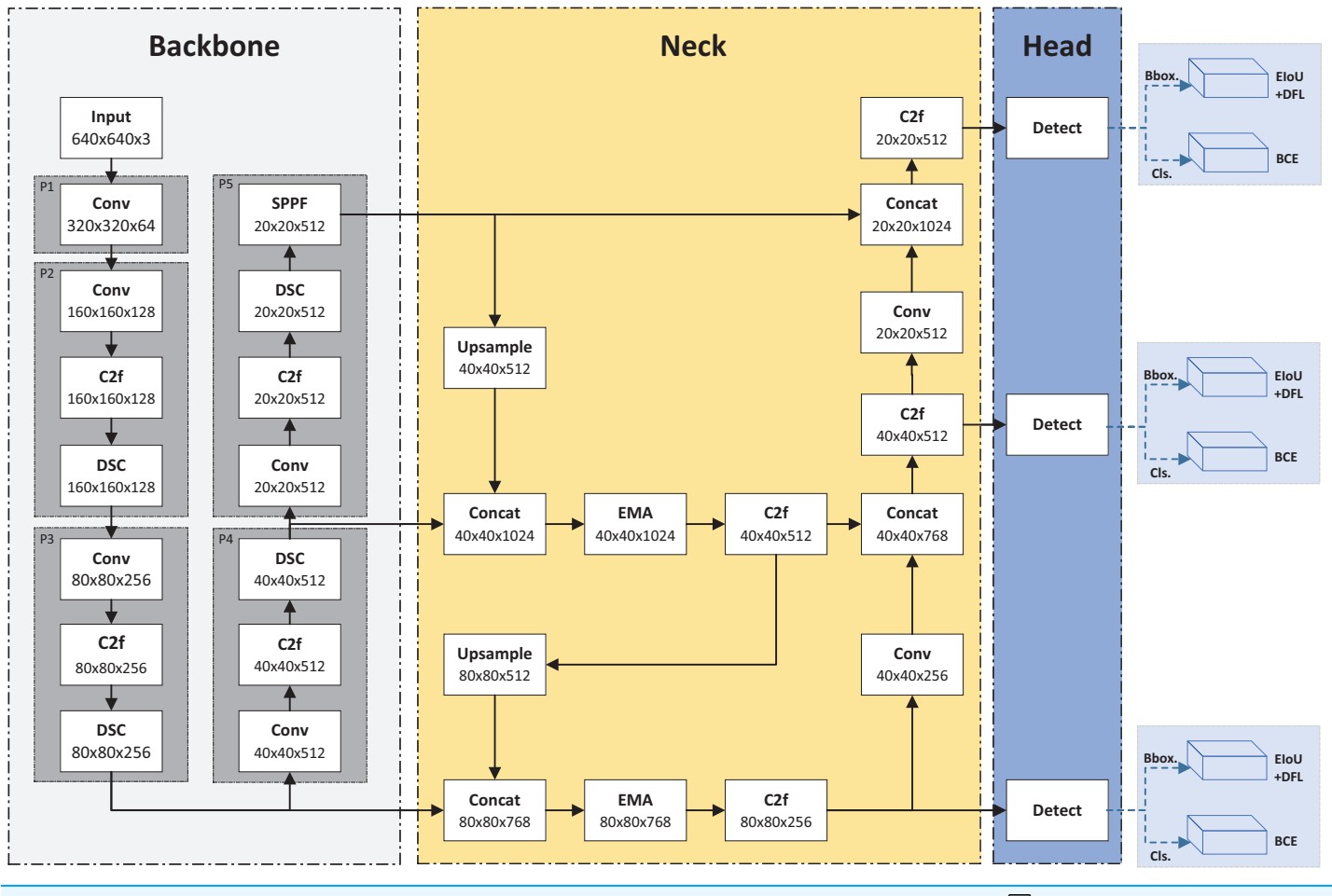

**Figure 1 Framework of YOLOv8-DEE.**

to each channel of the input feature map, then concatenates the outputs to form the final result. With *N* input channels, *N* separate convolution kernels produce *N* output channels, which are then concatenated to create the final output feature map with *N* channels. Pointwise convolution, a $1 \times 1$ convolution, serves two purposes in DSC: it allows the number of output channels to be adjusted freely and performs channel fusion on the feature map produced by depthwise convolution.

The DSC module performs a convolution operation on one feature map at a time and then uses a $1 \times 1$ convolution for output after fusion. This process significantly reduces computational complexity compared to standard convolution operations, with minimal loss of accuracy. Consequently, many studies adopt the DSC module as a replacement for traditional convolutional structures, making it an effective tool for developing lightweight networks (*Hung, Zhang & Jiang, 2019*; *Wang et al., 2020*).

Unlike most research, the proposed backbone in this article does not use DSC modules as replacements for traditional convolutions but integrates them directly into the backbone network. Specifically, DSC modules are added to layers P2 to P5. In layers P2 to P4, the

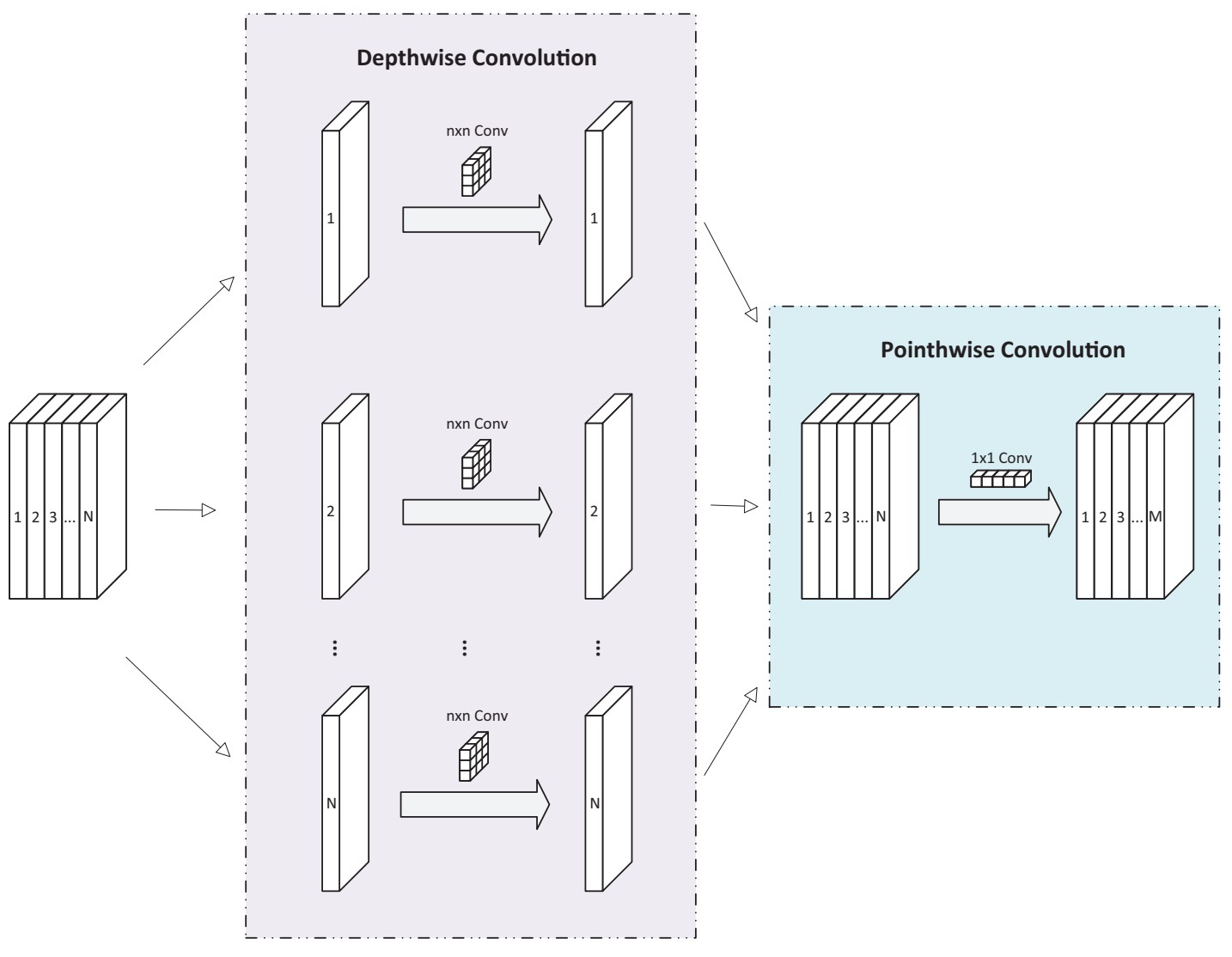

**Figure 2 Diagram of DSC module.**

DSC module follows the C2f module, while in P5, it is placed between the C2f and spatial pyramid pooling-fast (SPPF) modules. Experiments demonstrate that these modifications minimally increase the network's computational load while significantly enhancing its nonlinear expression capability, thereby improving the network's ability to fit complex functions and boosting detection performance.

## Enhanced neck with EMA module

The original YOLOv8 model uses the Path Aggregation Network (PANet) structure based on the FPN architecture to enhance feature extraction. Unlike the traditional FPN, PANet fuses both top-to-bottom and bottom-to-top feature maps. However, PCB images often contain complex backgrounds with many elements irrelevant to defects, such as text,

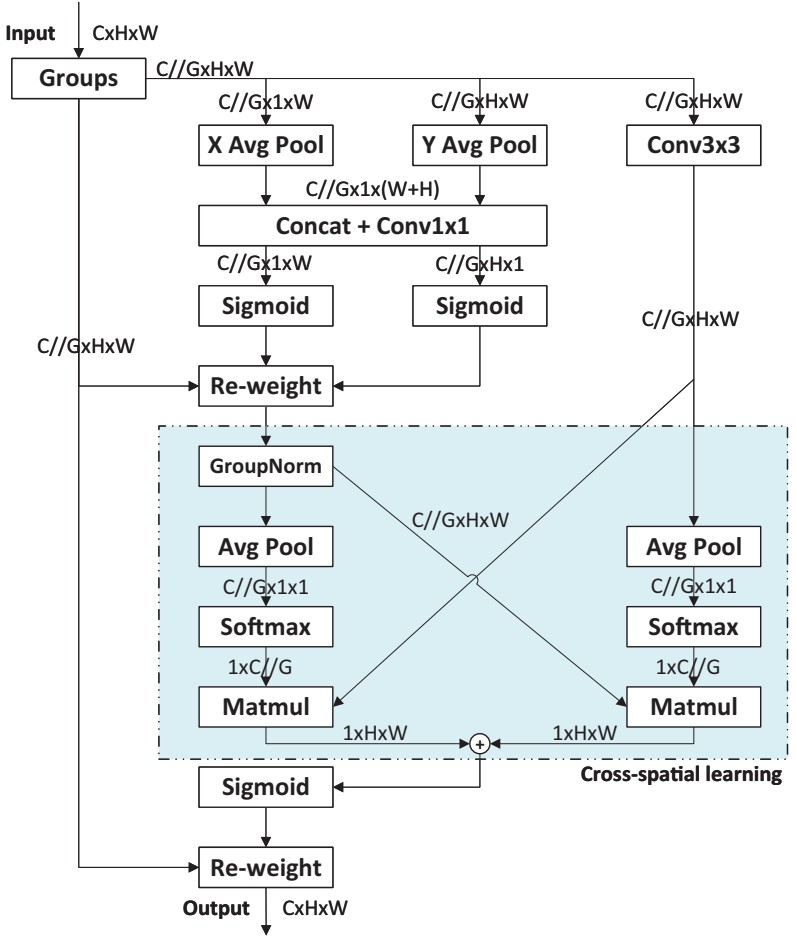

**Figure 3** Basic process of EMA module.

copper wires, mimeographs, and tin foil. The traditional neck structure and even PANet structure are insufficient to focus on defect features amid such cluttered background information. In other words, effectively integrating feature maps from different convolutional layers is crucial for designing enhanced feature extraction networks. To meet this challenge, an improved neck based on the efficient multi-scale attention (EMA) module (*Ouyang et al., 2023*) is proposed. Figure 3 presents the basic process of EMA module. The EMA module enhances interaction capabilities between different feature layers, improving the model's detection performance. Specially, EMA employs a cross-space learning method to establish short and long information dependencies *via* a multi-scale parallel sub-network. It reshapes some channels into batch dimensions and designs two mutually learning sub-networks to learn input features across spaces, finally fusing their outputs. Compared to other attention modules, EMA offers superior performance with lower computational overhead.

In enhanced neck design, two EMA modules are incorporated into the upsampling stage of the original PANet. The first EMA module is positioned after the Concat module, which merges the upsampled P5 feature map with the P4 feature map. This EMA module further

extracts high-dimensional feature information from the fused feature using an attention mechanism. The second EMA module is placed after another Concat module, which merges the P3 stage feature map with the upsampled features from the first EMA module. This module further extracts feature information from the upsampled features integrated with lower-level features.

## EIoU loss

The original YOLOv8 framework utilizes CIoU loss and distribution focal loss (DFL) technology for bounding box regression and binary cross-entropy (BCE) for classification prediction. CIoU loss serves as the loss function for target positioning, effectively measuring the model's accuracy in detecting the target's position and size, thereby enhancing performance evaluation. The formula for CIoU loss is as follows:

$$L_{CIoU} = 1 - IoU + \frac{\rho^2(b, b^{gt})}{c} + \alpha v \tag{1}$$

where $IoU$ denotes intersection over union, $b$ and $b^{gt}$ denote the centroids of the predicted bounding box and the true bounding box, respectively; $\rho$ denotes the Euclidean distance between the two centroids; $c$ denotes the diagonal length of the smallest closed rectangle containing the predicted bounding box and the true bounding box; $v$ is used to quantify the consistency of the aspect ratio; and $\alpha$ is a weight function. The equations for $\alpha$ and $v$ are as follows:

$$\alpha = \frac{v}{(1 - IoU) + v} \tag{2}$$

$$v = \frac{4}{\pi^2} \left( \arctan \frac{w_{gt}}{h_{gt}} - \arctan \frac{w}{h} \right) \tag{3}$$

where $w_{gt}$ and $h_{gt}$ denote the width and height of the ground-truth bounding box, and $w$ and $h$ denote the width and height of the prediction bounding box, respectively.

Although CIoU loss considers both the width-height ratio of the regression box and the distance between the centroids of the true and predicted bounding boxes, it has a limitation: it only uses the width-height ratio as an influencing factor. If two bounding boxes have the same center points and width-height ratio but different dimensions, CIoU loss may produce consistent results that do not align with the regression target. This deficiency becomes more pronounced in PCB defect detection tasks, where a large detection scale can exacerbate missed detections. To address this, EIoU loss (*Zhang et al., 2022*) was introduced. EIoU loss separates the influencing factors of the aspect ratio of the predicted and true bounding boxes, calculating their height and width independently. EIoU loss consists of three components: IoU loss ($L_{IoU}$), distance loss ($L_{dis}$), and height-width loss ($L_{asp}$), which account for the overlapping area, the distance between center points, and the aspect ratio. The height-width loss directly minimizes the difference in height and width between the predicted and true bounding boxes, enabling faster convergence and improved positioning accuracy. In addition to its use in PCB defect

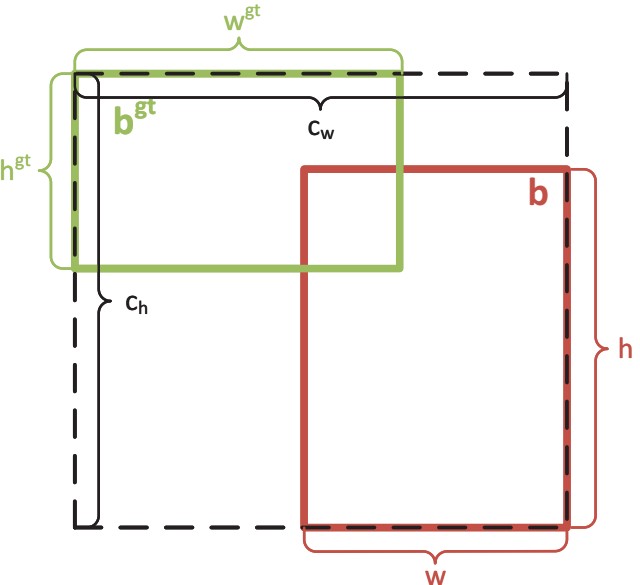

**Figure 4  Schematic diagram of EIoU loss.**

detection, the EIoU loss function has also been successfully applied in various other domains (*Jiao & Xing, 2024*; *Trinh et al., 2024*). The formula for EIoU loss is as follows:

$$L_{EIoU} = L_{IoU} + L_{dis} + L_{asp}$$
$$= 1 - IoU + \frac{\rho^2(b, b^{gt})}{(c_w)^2 + (c_h)^2} + \frac{\rho^2(w, w^{gt})}{(c_w)^2} + \frac{\rho^2(h, h^{gt})}{(c_h)^2} \qquad (4)$$

where $c_w$ and $c_h$ denote the width and height of the minimum enclosing box of ground-truth bounding box and prediction bounding box, respectively. Figure 4 shows the schematic diagram of EIoU loss.

# EXPERIMENT AND RESULTS

## Experimental configuration

The experiments are conducted on a high-performance computing environment. The system is equipped with an AMD EPYC 7742 64-Core processor and an NVIDIA A100-SXM4-40GB GPU, running Ubuntu 20.04.2 as the operating system. The deep learning models are implemented using PyTorch version 1.8.1 and Python 3.8.10. For optimal hardware utilization, CUDA version 12.2 is employed to accelerate GPU computation. During the training process, the batch size is set to 32, and the models are trained for 300 epochs to ensure sufficient learning and convergence of the algorithms.

## Dataset

### HRIPCB dataset

The public PCB defects dataset HRIPCB (*Huang et al., 2020*) from Peking University contains 693 images of six defect types, with an average pixel size of 2,777 × 2,188. Due to

the large size of the original images compared to the input size of typical object detection models, images are cropped into sub-images with $640 \times 640$ pixels size by using sliding window technique. To prevent target frames from being segmented during cropping, some sliding windows are adjusted appropriately. A total of 4,205 images are obtained after cropping, each containing 1–4 defects. Figure 5 introduces the six types of defects on the cropped HRIPCB dataset. The training set and test set are divided in a 9:1 ratio. The number of each defect category is shown in Table 1.

### DeepPCB dataset

All images on DeepPCB defect dataset (*Tang et al., 2019*) were obtained from a linear scan CCD. The original size of the test images is about $16k \times 16k$ pixels, and then they are cropped into many sub-images of size $640 \times 640$, a total of 1,500 pairs of template and tested images with annotations, including six common PCB defect types: open, short, mousebite, spur, pin-hole, spur. Figure 6 introduces the six types of defects on the DeepPCB dataset. The number of each defect category is shown in Table 2.

## Evaluation metrics

To evaluate the model's performance, the common metrics in object detection tasks are used in this article, which include precision (P), recall (R), F1 score (F1), average precision (AP), and mean average precision (mAP). The formulas for these metrics are as follows:

$$P = \frac{TP}{TP + FP} \tag{5}$$

$$R = \frac{TP}{TP + FN} \tag{6}$$

$$F1 = \frac{2 \cdot P \cdot R}{P + R} \tag{7}$$

$$AP = \int_0^1 p(r)dr \tag{8}$$

$$mAP = \frac{1}{N}\sum_{i=0}^{N} AP_i \tag{9}$$

where $TP$ represents the number of targets detected correctly; $FP$ represents the number of targets detected incorrectly; $FN$ represents the number of defect samples falsely detected; $P$ represents precision; $R$ represents recall; $AP$ values are the area enclosed under the curves of $P$ and $R$, $mAP$ represents the sum of AP values of all categories, and $N$ represents the total number of categories.

To further evaluate the model's accuracy at different IoU thresholds and target sizes, several performance metrics from the COCO dataset, including $AP_{0.5}$, $AP_{0.75}$, $AP_{0.5:0.95}$, $AP_S$, $AP_M$, and $AP_L$ are also used as evaluating indicators in the comparison experiments. $AP_{0.5}$ is the average precision for each category at an IoU threshold of 0.50, and $AP_{0.75}$ is the average precision at an IoU threshold of 0.75. $AP_{0.5:0.95}$ is the average precision across IoU thresholds from 0.50 to 0.95 in 0.05 increments. $AP_S$ represents the average precision

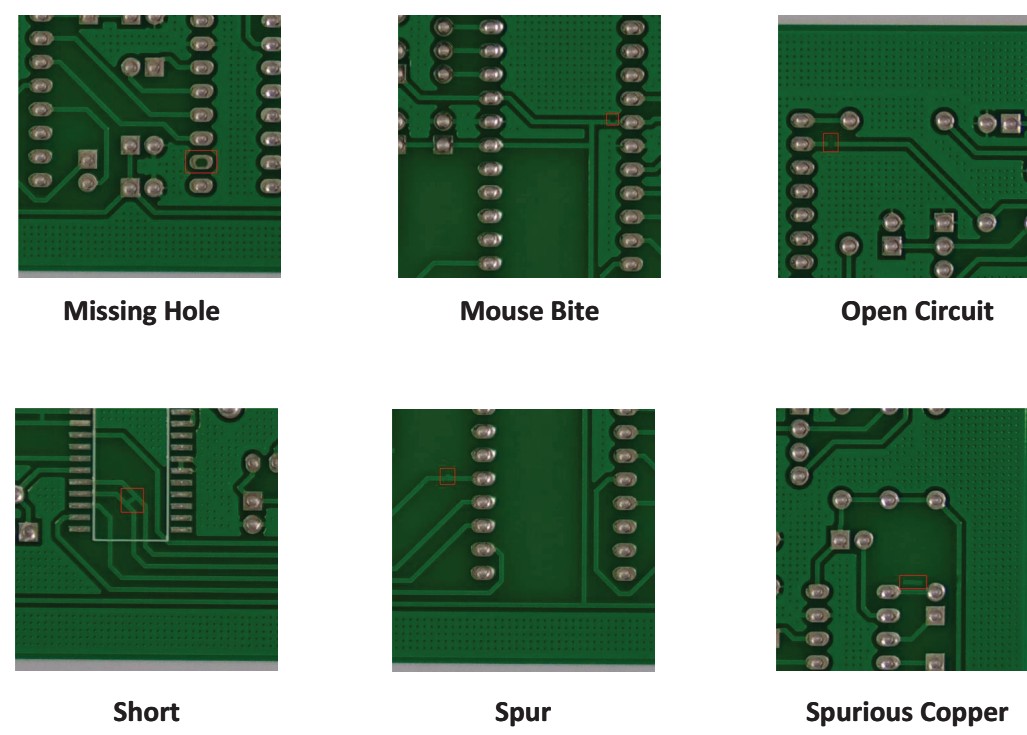

**Figure 5  Cropped HRIPCB dataset defect diagram.**

**Table 1  Number of each defect category on HRIPCB dataset.**

| Category | Numbers |
| --- | --- |
| Missing hole | 748 |
| Mouse bite | 756 |
| Open circuit | 706 |
| Short | 782 |
| Spur | 741 |
| Spurious copper | 772 |

for small objects (area $<32 \times 32$), $AP_M$ for medium objects (area between $32 \times 32$ and $96 \times 96$), and $AP_L$ for large objects (area $>96 \times 96$).

## Experiment result
### Ablation results

To evaluate the impact of the improvements on the YOLOv8-L model, ablation experiments are conducted on the three enhancements (Backbone, Neck, Loss) using the cropped HRIPCB dataset. Table 3 presents the results of these experiments based on the original YOLOv8-L model and its improved components. The base YOLOv8-L model achieved a commendable precision of 96.64% and a recall of 94.22%. Integrating DSC alone boosted recall to an impressive 96.00%, the highest among individual enhancements, and improved AP across all scales, demonstrating enhanced effectiveness in detecting objects of varying sizes.

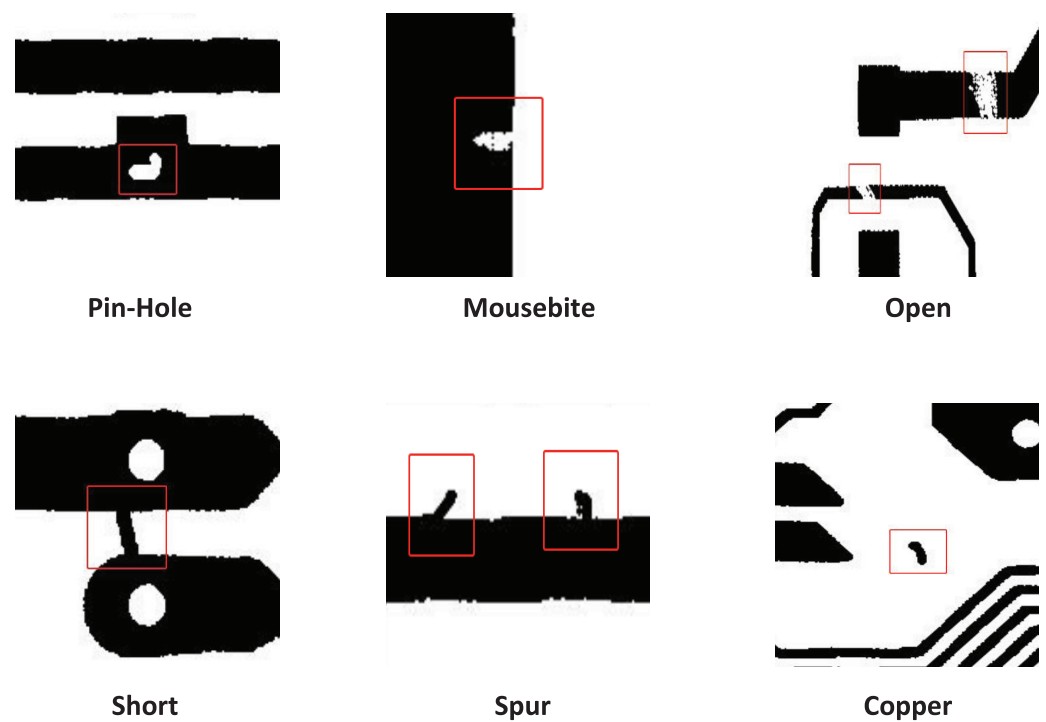

**Figure 6** DeepPCB dataset defect diagram.

**Table 2 Number of each defect category on the DeepPCB dataset.**

| Category | Numbers |
| --- | --- |
| Pin-hole | 1,320 |
| Mousebite | 1,748 |
| Open | 1,702 |
| Short | 1,317 |
| Spur | 1,445 |
| Copper | 1,321 |

**Table 3 Ablation experiment results.** The bold entries show the best results of different models under this evaluation metric.

| Model | P | R | $AP_{0.5:0.95}$ | $AP_{0.5}$ | $AP_S$ | $AP_M$ | $AP_L$ |
| --- | --- | --- | --- | --- | --- | --- | --- |
| YOLOv8-L | 96.64% | 94.22% | 52.6% | 95.0% | 34.5% | 52.4% | 60.2% |
| YOLOv8 + DSC | 96.70% | **96.00%** | 59.2% | 96.5% | 63.8% | 58.9% | 66.5% |
| YOLOv8 + EMA | 96.88% | 95.80% | 58.7% | 97.0% | 61.3% | 58.6% | 67.7% |
| YOLOv8 + EIoU | 97.30% | 95.01% | 60.2% | 97.0% | 64.7% | 60.0% | 68.9% |
| YOLOv8 + DSC + EMA | 96.86% | 95.80% | 58.4% | 96.7% | 57.5% | 58.2% | 67.3% |
| YOLOv8 + DSC + EIoU | 97.03% | 94.84% | 60.2% | 96.6% | 59.0% | 60.1% | 65.8% |
| YOLOv8 + EMA + EIoU | 97.59% | 93.45% | 59.7% | 97.3% | 57.7% | 59.6% | 64.7% |
| **YOLOv8-DEE** | **97.65%** | 95.43% | **60.4%** | **97.4%** | **69.5%** | **60.2%** | **71.1%** |

**Table 4 Compare experiment results with YOLOv8-L baseline model.** The bold entries show the best detection results of the defect type by the YOLOv8-L baseline model and our proposed YOLOv8-DEE model under the same evaluation metric.

| Category | YOLOv8-L | | | | YOLOv8-DEE | | | |
|---|---|---|---|---|---|---|---|---|
| | P | R | F1 | AP | P | R | F1 | AP |
| Missing hole | **96.85%** | **99.19%** | **98.01%** | **99.00%** | 96.34% | 98.75% | 97.53% | 98.31% |
| Mouse bite | 95.83% | 94.52% | 95.17% | 95.73% | **97.59%** | **96.43%** | **97.01%** | **99.13%** |
| Open circuit | 97.03% | 95.15% | 96.08% | 97.40% | **100.00%** | **97.53%** | **98.75%** | **98.74%** |
| Short | 96.00% | **96.00%** | **96.00%** | 94.39% | **96.51%** | 94.32% | 95.40% | **97.17%** |
| Spur | 98.32% | 86.03% | 91.76% | 92.26% | **100.00%** | **86.75%** | **92.90%** | **96.23%** |
| Spurious copper | **95.80%** | 94.48% | 95.14% | 94.89% | 95.45% | **98.82%** | **97.11%** | **99.30%** |
| Average | 96.64% | 94.22% | 95.36% | 95.61% | **97.65%** | **95.43%** | **96.45%** | **98.15%** |

Adding the EMA module slightly increased precision to 96.88% and improved AP for medium ($AP_M$) and large ($AP_L$) objects, indicating its effectiveness in handling different object scales. EIoU, focused on refining bounding box accuracy, raised precision to 97.30% and significantly enhanced performance for larger objects, achieving an AP of 68.9%.

However, when DSC and EMA are combined, their individual contributions to improving feature extraction may overlap in terms of spatial focus. DSC's ability to reduce the model's complexity while retaining spatial information might already capture the same aspects that EMA is designed to enhance. As a result, the combined effect does not lead to a significantly higher improvement compared to when each module is applied individually, suggesting that they may be optimizing similar aspects of the feature map. In contrast, pairing DSC and EIoU achieved an AP of 60.2%, indicating that structural model changes and bounding box accuracy enhancements complement each other well.

The final model, YOLOv8-DEE, which incorporates all three enhancements–DSC, EMA, and EIoU–outperforms all others. This model achieves the highest metrics across most categories, excelling particularly in detecting small objects ($AP_S$ at 69.5%) and large objects ($AP_L$ at 71.1%). This comprehensive enhancement strategy demonstrates how simultaneous architectural and functional upgrades can substantially elevate a detection model's performance, especially under varying object detection challenges.

### Comparison experiment with YOLOv8-L

To further verify the effectiveness of the model, a comparation experiment is conducted between the YOLOv8 baseline model and the enhanced YOLOv8-DEE on different types of defects on the cropped HRIPCB dataset. The experimental results are shown in Table 4. On average, YOLOv8-DEE outperforms YOLOv8-L across all metrics, including precision, recall, F1-score, and AP.

In the 'Mouse Bite' category, YOLOv8-DEE improves precision by nearly two percentage points and recall by almost two percentage points, resulting in a higher F1-score and a substantial boost in AP. This indicates that YOLOv8-DEE better detects irregular and less distinct features. For 'Missing Hole', the baseline model slightly

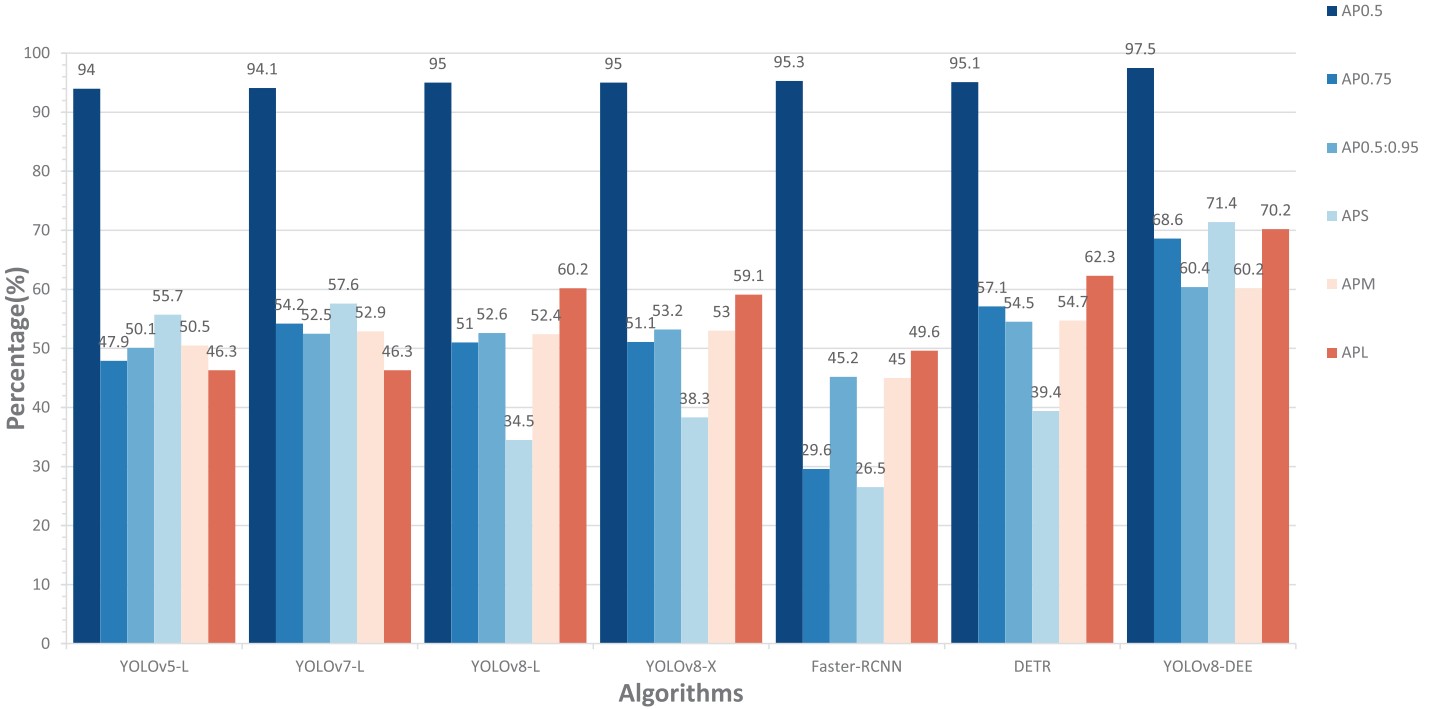

**Figure 7 Comparisons between YOLOv8-DEE and state-of-the-arts on the HRIPCB dataset.**

outperforms YOLOv8-DEE across all metrics, suggesting that for highly predictable defects with consistent features, the baseline model may be sufficiently robust.

In the 'Open Circuit' category, YOLOv8-DEE achieves perfect precision and significantly improves recall and AP, showcasing its superior capability to identify open circuit defects, which are often challenging due to their fine and dispersed nature. For 'Short' defects, both models perform comparably in precision, but YOLOv8-DEE shows a marginal improvement in AP, indicating slight enhancements in detecting and delineating connected circuit paths.

In the 'Spur' category, YOLOv8-DEE achieves perfect precision and slightly higher recall, significantly improving both F1-score and AP. For 'Spurious Copper', recall increases remarkably from 94.48% to 98.82% with YOLOv8-DEE, along with corresponding rises in F1-score and AP, which are crucial for detecting random, unexpected copper deposits.

These results clearly demonstrate that YOLOv8-DEE provides a more robust solution for detecting diverse PCB defects compared to the baseline model, particularly in categories that benefit from enhanced attention mechanisms and improved bounding box accuracy. The consistent improvements across most categories suggest that the integrated approach of YOLOv8-DEE effectively enhances the model's overall performance, validating the incorporation of these advanced modules into the detection framework.

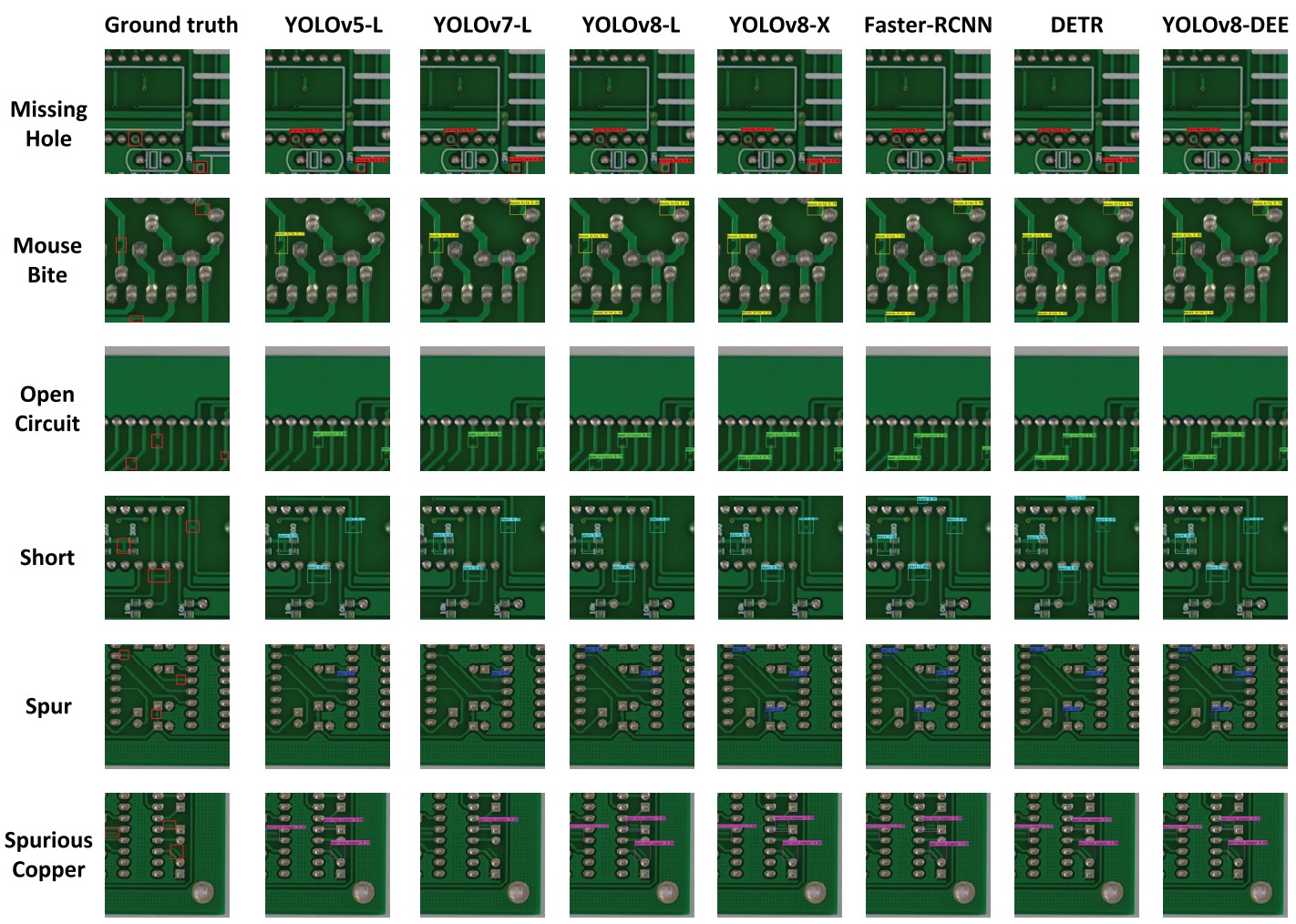

|  | Ground truth | YOLOv5-L | YOLOv7-L | YOLOv8-L | YOLOv8-X | Faster-RCNN | DETR | YOLOv8-DEE |

**Missing Hole**

**Mouse Bite**

**Open Circuit**

**Short**

**Spur**

**Spurious Copper**

**Figure 8  Detection effect on different models.**               

### Comparison experiment on the cropped HRIPCB dataset

In addition, this study compared the performance of our model with other state-of-the-art models, including YOLOv5-L, YOLOv7-L (*Wang, Bochkovskiy & Liao, 2023*), YOLOv8-X, Faster-RCNN, and DETR (*Carion et al., 2020*) on cropped HRIPCB dataset. The comparative performance of these models, including the proposed YOLOv8-DEE, is detailed in Fig. 7. This analysis focuses on several key metrics: AP at different IoU thresholds ($AP_{0.5}$, $AP_{0.75}$, $AP_{0.5:0.95}$), and average precision for small ($AP_S$), medium ($AP_M$), and large ($AP_L$) objects. The YOLOv8-DEE model demonstrates a clear advantage across all specified metrics, underscoring its robustness and efficiency in object detection tasks across varying scales and complexities.

In AP metrics at different IoU thresholds, the proposed YOLOv8-DEE model significantly outperforms all other models, achieving an $AP_{0.5}$ of 97.5%. The closest competitors, YOLOv5-L and Faster-RCNN, show competitive but lower performance.

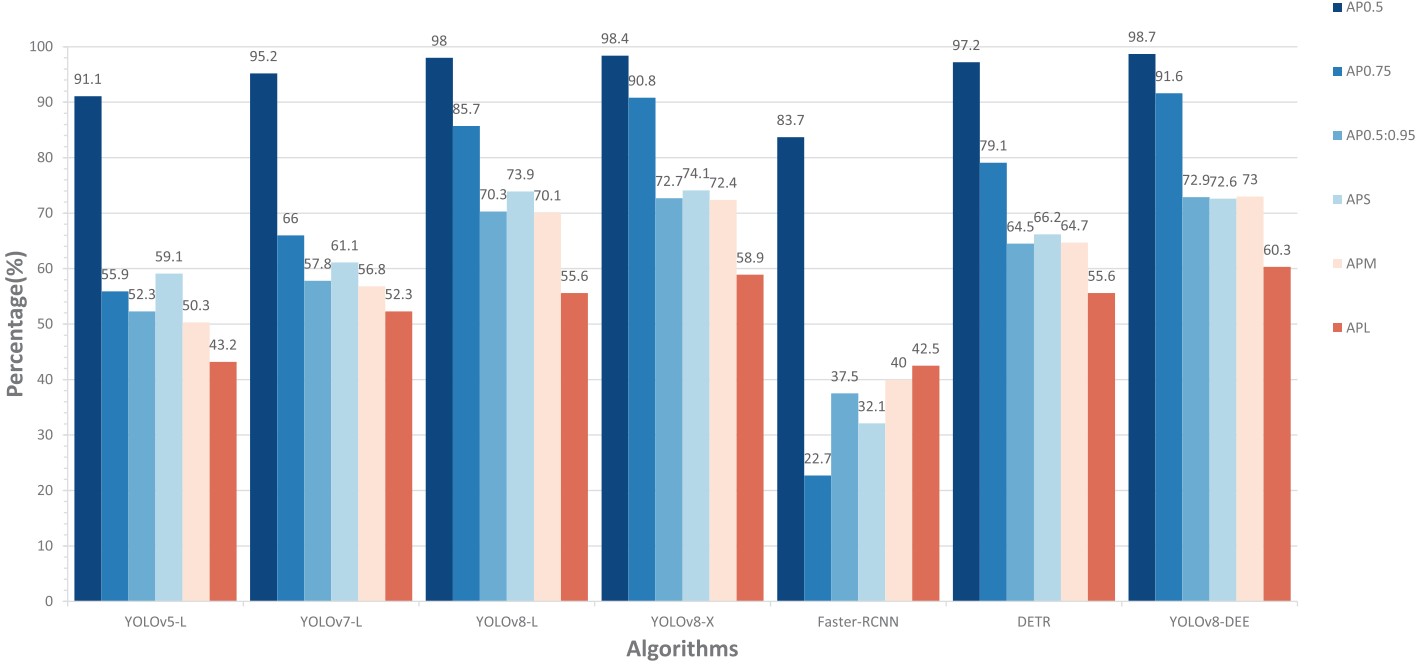

**Figure 9 Comparisons between YOLOv8-DEE and state-of-the-arts on the DeepPCB dataset.**

Additionally, YOLOv8-DEE leads in more stringent metrics, with $AP_{0.75}$ at 68.6% and $AP_{0.5:0.95}$ at 60.4%. This indicates that YOLOv8-DEE not only identifies objects more accurately but also aligns better with the ground truth, a critical factor in applications requiring high precision. The visualization results of detection effect on these models are displayed in Fig. 8. It can be clearly seen that PCB defects only account for a small proportion of the board. YOLOv8-DEE can detect all defect types and obtain more accurate defect locations than other models.

Regarding performance across object sizes, YOLOv8-DEE demonstrates a remarkable ability to detect small objects, achieving an $AP_S$ of 71.4%, significantly higher than any other model. This performance is crucial for scenarios where small object detection is essential, such as PCB defect detection, medical imaging, and satellite imagery. For medium-sized objects ($AP_M$), while the improvements are less dramatic, YOLOv8-DEE still matches the best of the other models, showing balanced performance across object sizes. Furthermore, for large objects, YOLOv8-DEE excels with an $AP_L$ of 70.2%, indicating robust performance in detecting larger objects, which is often challenging due to higher variability in appearance.

### Comparison experiment on DeepPCB dataset

To further validated the performance of YOLOv8-DEE, a comparion experiment is conducted on the DeepPCB dataset. Figure 9 presents the comparison of the experimental results for various models, including YOLOv5-L, YOLOv7-L, YOLOv8-L, YOLOv8-X,

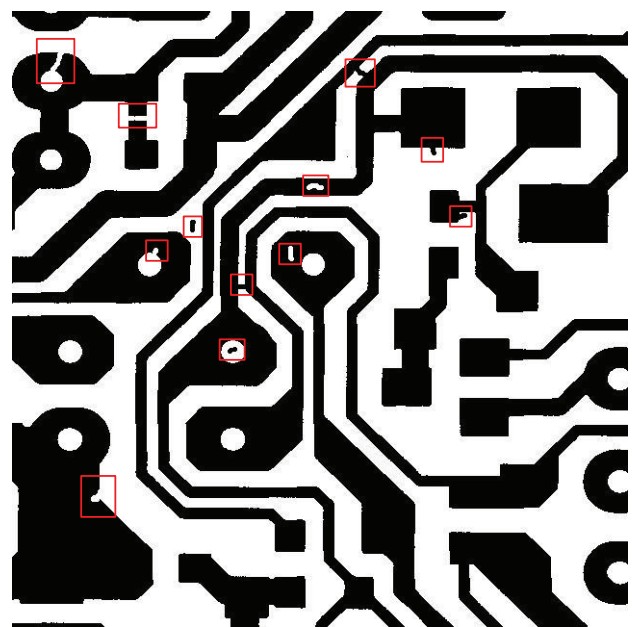
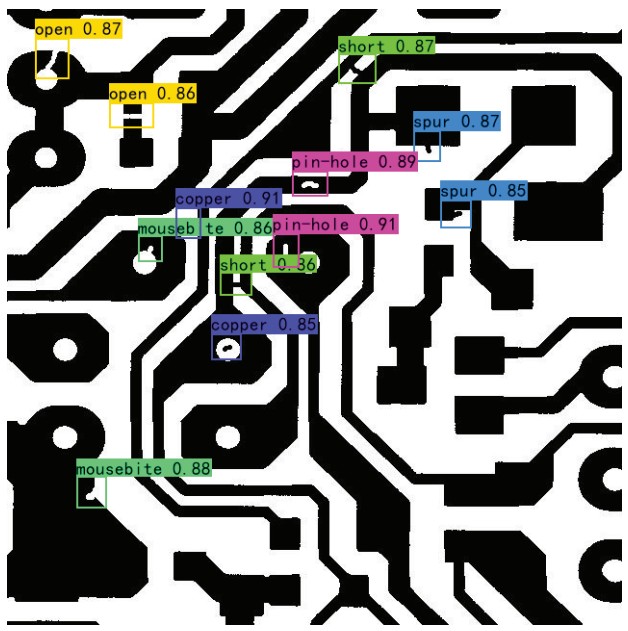

**Ground Truth**  **YOLOv8-DEE**

**Figure 10 Visual result of YOLOv8-DEE on the DeepPCB dataset.**

Faster-RCNN, DETR, and the proposed YOLOv8-DEE. The evaluation metrics used are $AP_{0.5}$, $AP_{0.75}$, $AP_{0.5:0.95}$, $AP_S$, $AP_M$, and $AP_L$.

In terms of $AP_{0.5}$, the YOLOv8-DEE achieved the highest score at 98.7%, outperforming all other models. Although YOLOv8-X has a more complex feature extraction network, its detection performance is still slightly lower than the proposed model. On the other hand, all anchor-based models (YOLOv5-L, YOLOv7-L,Faster-RCNN) perform significantly worse than anchor-free models (YOLOv8-L, YOLOv8-X, DETR, YOLOv8-DEE), especially Faster-RCNN, which achieves the lowest detection accuracy on all detection metrics. This indicates that anchor-free type detection heads may be more suitable for detecting multi-scale PCB defects. The visual result of YOLOv8-DEE on DeepPCB is shown in Fig. 10.

Regarding the AP metrics of different sizes, the YOLOv8 series models have achieved a greater advantage compared to other detection models. The proposed model shows stronger performance in detecting small object defects and large object defects compared to YOLOv8-L and YOLOv8-X.

## CONCLUSIONS

To address the challenges of insufficient detection accuracy and complex backgrounds in PCB defect detection, this article proposes the YOLOv8-DEE algorithm, an enhancement of YOLOv8-L. The integration of DSC, EMA, and EIoU modules in this model significantly improves performance compared to existing architectures such as YOLOv5-L,

YOLOv7-L, YOLOv8-X, Faster R-CNN, and DETR. Through extensive experiments on the HRIPCB and DeepPCB datasets, YOLOv8-DEE demonstrates superior performance, achieving mAP scores of 97.5% and 98.7%, respectively. These results outperform state-of-the-art methods, particularly in the detection of both small and large PCB defects. Moreover, ablation studies verify the robustness of YOLOv8-DEE. However, the algorithm lacks lightweight processing techniques such as cropping and knowledge distillation, resulting in a relatively large number of parameters and no significant improvement in inference time. Future optimization efforts should focus on reducing the model's size and computational demands without compromising detection accuracy.

### Funding
This work was supported by the Research Projects of Department of Education of Guangdong Province (2023KCXTD077). The funders had no role in study design, data collection and analysis, decision to publish, or preparation of the manuscript.

### Grant Disclosures
The following grant information was disclosed by the authors:
Department of Education of Guangdong Province: 2023KCXTD077.

### Competing Interests
Fakhrozi Che Ani and Zol Effendi Zolkefli are employed by Western Digital SanDisk Storage Malaysia.

### Author Contributions
- Feifan Yi conceived and designed the experiments, performed the experiments, analyzed the data, performed the computation work, prepared figures and/or tables, authored or reviewed drafts of the article, and approved the final draft.
- Ahmad Sufril Azlan Mohamed conceived and designed the experiments, analyzed the data, performed the computation work, prepared figures and/or tables, authored or reviewed drafts of the article, and approved the final draft.
- Mohd Halim Mohd Noor conceived and designed the experiments, analyzed the data, prepared figures and/or tables, authored or reviewed drafts of the article, and approved the final draft.
- Fakhrozi Che Ani performed the experiments, authored or reviewed drafts of the article, and approved the final draft.
- Zol Effendi Zolkefli performed the experiments, authored or reviewed drafts of the article, and approved the final draft.

### Data Availability
The DeepPCB dataset is available at arXiv: Sanli Tang, Fan He, Xiaolin Huang, Jie Yang, 2019. Online PCB Defect Detector On A New PCB Defect Dataset. https://doi.org/10.48550/arXiv.1902.06197.

The HRI PKU-Market-PCB dataset is available at: https://robotics.pkusz.edu.cn/resources/dataset.

## Supplemental Information

Supplemental information for this article can be found online at http://dx.doi.org/10.7717/peerj-cs.2548#supplemental-information.

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
