# Peer review of "YOLOv8-DEE: a high-precision model for printed circuit board defect detection"

_PeerJ Computer Science, doi:10.7717/peerj-cs.2548_

## Round 0.1 · original submission · Major Revisions

Dear authors,
You are advised to critically respond to all comments point by point when preparing an updated version of the manuscript and while preparing for the rebuttal letter. Please address all comments/suggestions provided by reviewers, considering that these should be added to the new version of the manuscript.

Kind regards,
PCoelho

Reviewer 1 ·

Basic reporting

English is clear and readable for nonnative speakers. The manuscript uses appropriate terminology. In manuscript were investigated public available datasets of PCB defect (raw data) e.g. HRIPCB and DeepPCB. Background is sufficient. Figures underline method properties and are relevant, high quality, well labelled & described.

Experimental design

Significant challenges and contributions of the manuscript are underlined with precision at pages 1-2. The proposed model YOLOv8-DEE is verified in sub-images of size 640×640 obtained from DeepPCB and HRIPCB after preprocessing described in the article. The hardware setup is given in Table 1. The proposed model is compared with other variants of the YOLOv8 model in terms of metrics, e.g. precision, recall, F1-score. The model is described in an appropriate manner.

Validity of the findings

The improvement of the YOLOv8 model is clearly given. Conducted experimental validation of model provides benefits of enhanced model in terms of metrics e.g. precision, recall, F1-score. Conclusions are appropriate.

Additional comments

Dear authors,
The proposed manuscript addresses an important topic of fault localisation at printed circuit boards PCBs. PCBs are inspected with image sensor (colour camera) and additional white-light source. Collected images are used as input to the method of fault localisation at image. Authors use method of You Only Look Once YOLOv8 with enhancement to YOLOv8-DSC-EMA-EIoU (YOLOv8-DEE).

Cite this review as

Reviewer 2 ·

Basic reporting

no comments

Experimental design

The paper lacks a description of existing defects of printed circuit boards. What are the main types of defects? Which defects are the most difficult to detect, which are more common? In the results, the authors show an example of 6 types of defects, but how many are there in total, and why were these chosen?

Validity of the findings

The "Conclusions" section should include numerical results of comparative experiments, as well as a description of both the advantages of the proposed solution and its limitations.

Additional comments

Figure 4. The authors can reduce Figure 4 so that it does not take up too much space.

Cite this review as

·

Basic reporting

Overall, it is a great work that proposed a novel YOLO-V8-based algorithm for defect detection in PCB.
However, there are some writing issues need to be addressed:
1. In the introduction, more background information related to quality control in PCB inspection and the types of common PCB defects would be helpful.
2. Please provide more details to explain one-stage and two-stage algorithms.
3. Other than the template-based method, are there any other traditional ML methods applied for PCB detection? It would be great to add 1-2 other methods with literature reviews in this section.
4. There are some weird ways to cite work, such as:
“the authors of Long et al. (2023) present an improved YOLOv8-based algorithm for PCB defect detection. ”; Please check others and correct the citation expression.
5. Please provide the full name of abbreviations when first mentioning, such as : CSP, FPN, SPPF, etc…
6. What is the “COCO dataset”, and what’s the difference between that dataset and this application?

Experimental design

The experiment design is good.

Validity of the findings

Overall, the results are good enough to show the efficiency of proposed algorithm,
But the Results in Figure 8 make it hard to see what exactly has been recognized, it will be beneficial to enlarge some of them to show the results.

Reviewer 4 ·

Basic reporting

This paper proposes improvements on YOLOv8 architecture by incorporating two modules in the backbone and the neck of the network, while examining a different loss function than the original.
The proposed method is sufficiently explained and validated and in two PCB datasets, where it is demonstrated through an ablation process that improvement is achieved by all the proposed modifications. However, there are a few issues that should be addressed to improve the quality of the work:
1. Basic reporting
1.1. While the language of the manuscript is generally adequate, several sections should be revised to maintain a more formal scientific tone and improve clarity. For example, phrases such as “Detecting such small defects is extremely difficult,” “may lead to complete PCB function loss,” “extreme pursuit of inspection accuracy,” and “iconic one-stage detection model”, “slide windows technology” should be rephrased.
1.2. Please avoid using subjective or vague terms that could be open to interpretation, such as “excellent results” and “tiny size.” Quantitative descriptions would be more precise and helpful for readers.
1.3. For clarity, please provide the full names of the CIOU and EIOU loss functions when first mentioned, as this will help readers unfamiliar with these terms.
1.4. In the Introduction, while the challenges of PCB defect detection are addressed, the descriptions should be revised for greater clarity and precision. For instance, in the first challenge, associating tiny defects with the number of pixels they occupy in a high-resolution image does not fully convey the complexity of the challenge, as it is also influenced by factors such as the accuracy and type of sensor used for defect detection. For example, in “Evangelidis, A., et al. (2023)[1] . the authors investigate the use of 3D laser technology to capture micro-scale defects in microelectronics and explore various sensors and networks aimed at reducing acquisition time and improving inspection speed. I suggest that each challenge be further analyzed in the related works section by expanding the current discussion. This will provide readers with a deeper understanding of the problem's background and offer insight into the current state of the literature on these specific challenges (see also comment ‘1.7’ below).
1.5. Furthermore, describing defects as “tiny” does not convey the specific physical scale relevant to PCBs, as small defects are common in various other manufacturing fields. Additionally, the term “Lightweight” could be replaced with a clearer explanation of the need for real-time or near-real-time detection in production environments. The challenge of “Precision” is also not clearly defined, as precision is crucial across all defect detection tasks in manufacturing. The use of “precision” and “accuracy” in this context may lead to confusion, as these terms are typically associated with evaluation metrics. Clarifying the specific challenge would avoid misinterpretation.
1.6. In the fourth paragraph of the Introduction, the manuscript presents a brief categorization of deep learning approaches. This section would be better placed earlier in the Introduction, ideally before the discussion of challenges, to provide a more logical flow to the manuscript.
1.7. The first paragraph of the "Deep Learning Based Methods for PCB Defect Detection" section repeats information about one-stage and two-stage networks, which has already been introduced earlier. Furthermore, this section does not clearly cover all the challenges in PCB manufacturing, presented in the introduction, as addressed by related works.
1.8. In ‘Overall Framework of YOLOv8-DEE’, it is suggested that “YOLOv8 has demonstrated impressive results on the COCO dataset Lin et al. (2014), its generalization across different datasets requires further investigation”. However, there are works in the literature that have already examined the latest versions of YOLO in different application domains, for instance in insulator defect detection (Jiang, T. et al. 2024)[2], in battery inspection (Tzelepakis, A. et al. 2023) [3], in road defect detection (Wang, J. et al. 2024) [4] , steel surface inspection (You, C. et al. 2024) [5] among others. Including these works would help the readers understand the network’s ability across different domains, while it will enhance self-containment in the paper.
1.9. The EIoU loss function has been examined in works of other domains also, thus it would be beneficial to add in ‘EIoU Loss’ section, a reference of how it has been already successfully used in literature in real-world applications in various domains, for instance in Jiao, Y. et al (2024) [6] or in Trinh et al. (2024) [7] .
1.10. Please consider reducing the size of Figure 2 and Figure 4, as their larger format does not provide additional information and a smaller version would be more concise without sacrificing clarity
1.11. Several paragraphs and subsections in the manuscript are too short, which impacts readability (e.g. “Experimental Configuration” includes only one sentence, or the paragraph in line 353), Please consider revising the paragraph structure by combining related ideas or extending existing paragraphs, to create more cohesive and fluid sections.

[1] Evangelidis, A., Dimitriou, N., Leontaris, L., Ioannidis, D., Tinker, G., & Tzovaras, D. (2023). A Deep Regression Framework Toward Laboratory Accuracy in the Shop Floor of Microelectronics. In IEEE Transactions on Industrial Informatics (Vol. 19, Issue 3, pp. 2652–2661). Institute of Electrical and Electronics Engineers (IEEE). https://doi.org/10.1109/tii.2022.3182343.
[2] Jiang, T., Hou, X. & Wang, M. Insulator Defect Detection Based on the CDDCR–YOLOv8 Algorithm. Int J Comput Intell Syst 17, 245 (2024). https://doi.org/10.1007/s44196-024-00654-x.
[3]Tzelepakis, A., Leontaris, L., Dimitriou, N., Koukidou, E., Bollas, D., Karamanidis, A., & Tzovaras, D. (2023). Automated Defect Detection In Battery Line Assembly Via Deep Learning Analysis. In 10th ECCOMAS Thematic Conference on Smart Structures and Materials. 10th ECCOMAS Thematic Conference on Smart Structures and Materials. Dept. of Mechanical Engineering & Aeronautics University of Patras. https://doi.org/10.7712/150123.9868.444540.
[4] Wang, J., Meng, R., Huang, Y., Zhou, L., Huo, L., Qiao, Z., & Niu, C. (2024). Road defect detection based on improved YOLOv8s model. In Scientific Reports (Vol. 14, Issue 1). Springer Science and Business Media LLC. https://doi.org/10.1038/s41598-024-67953-3
[5] You, C., & Kong, H. (2024). Improved Steel Surface Defect Detection Algorithm Based on YOLOv8. In IEEE Access (Vol. 12, pp. 99570–99577). Institute of Electrical and Electronics Engineers (IEEE). https://doi.org/10.1109/access.2024.3429555.
[6] Jiao, Y., & Xing, L. (2024). Vehicle Target Detection Research Based on Enhanced YOLOv8. In 2024 4th International Conference on Neural Networks, Information and Communication (NNICE) (pp. 1427–1432). 2024 4th International Conference on Neural Networks, Information and Communication (NNICE). IEEE. https://doi.org/10.1109/nnice61279.2024.10498766
[7] Trinh, D. C., Mac, A. T., Dang, K. G., Nguyen, H. T., Nguyen, H. T., & Bui, T. D. (2024). Alpha-EIOU-YOLOv8: An Improved Algorithm for Rice Leaf Disease Detection. In AgriEngineering (Vol. 6, Issue 1, pp. 302–317). MDPI AG. https://doi.org/10.3390/agriengineering6010018


Experimental design

2.1. Could you provide a more thorough explanation of “overlapping functionalities” for DSC and EMA combination?
2.2. In Figure 8, it would be helpful to include the ground truth alongside the input image for better comparison. Additionally, the marked detections are not clearly visible; please improve the figure's quality to enhance visibility.
2.3. In Figure 9 and Figure 10, the caption should be corrected (“compare experiment results).
2.4. Visualization results should also be provided for the second datasets, DeepPCB, since only the quantitative results are presented.
2.5. Please avoid using subjective language such as “leading solution in object detection”, as it can detract from the objectivity and scientific rigor of the study.

Validity of the findings

The experimental findings are adequately explained and the proposed method is validated in two well-known PCB datasets from the literature, showing better performance than existing methods.

Cite this review as

---

## Round 0.2 · accepted · Accept

Dear authors, we are pleased to verify that you meet the reviewer's valuable feedback to improve your research.

Thank you for considering PeerJ Computer Science and submitting your work.

Kind regards
PCoelho

Reviewer 1 ·

Basic reporting

English is clear and readable for nonnative speakers. The manuscript uses appropriate terminology. In manuscript were investigated public available datasets of PCB defect (raw data) e.g. HRIPCB and DeepPCB. Background is sufficient. Figures underline method properties and are relevant, high quality, well labelled & described.

Experimental design

Significant challenges and contributions of the manuscript are underlined with precision at pages 1-2. The proposed model YOLOv8-DEE is verified in sub-images of size 640×640 obtained from DeepPCB and HRIPCB after preprocessing described in the article. The hardware setup is given in Table 1. The proposed model is compared with other variants of the YOLOv8 model in terms of metrics, e.g. precision, recall, F1-score. The model is described in an appropriate manner.

Validity of the findings

The improvement of the YOLOv8 model is clearly given. Conducted experimental validation of model provides benefits of enhanced model in terms of metrics e.g. precision, recall, F1-score. Conclusions are appropriate.

Additional comments

Dear authors,
The proposed manuscript addresses an important topic of fault localisation at printed circuit boards PCBs. PCBs are inspected with image sensor (colour camera) and additional white-light source. Collected images are used as input to the method of fault localisation at image. Authors use method of You Only Look Once YOLOv8 with enhancement to YOLOv8-DSC-EMA-EIoU (YOLOv8-DEE).

Cite this review as

Reviewer 2 ·

Basic reporting

The authors took into account all the comments, the article looks much better. The Figures look good, the structure of the paper has become much clearer.
I believe that this work is ready for publication and will be an excellent contribution to PeerJ.

Experimental design

no comments

Validity of the findings

The conclusions include numerical results of the study.

Cite this review as

·

Basic reporting

All the questions have been appropriately addressed. It is a good work to be published.

Experimental design

All the questions have been appropriately addressed. It is a good work to be published.

Validity of the findings

All the questions have been appropriately addressed. It is a good work to be published.

Additional comments

All the questions have been appropriately addressed. It is a good work to be published.

Reviewer 4 ·

Basic reporting

The authors have addressed all the suggested issues comprehensively, correcting language errors, improving literature review and presentation issues.

Experimental design

The authors have addressed all the issues raised in the review process. They have revised unclear figures, clarified ambiguous passages, and provided an additional figure to enhance understanding.

Validity of the findings

The authors have generally improved the presentation of their findings, elaborating on the raised issues.

Cite this review as